# The Effect of Different Environmental Factors on Milk Yield Characteristics of Holstein Fresian Cattle Raised with Different Production Scale on Teke Region of Turkey

**Cevat Sipahi**

Department of Livestock Economics and Management, Faculty of Veterinary, Mehmet Akif Ersoy University, 15030 Burdur, Turkey; cevatsipahi@mehmetakif.edu.tr; Tel.: +90-533-3695330

**Abstract:** This study was intended to determine whether there was any difference between the parameters of herd size and milk yield based on the hypothesis that the dairy cattle enterprises in the Teke Region used different production methods depending on their herd size. Total milk yield and 305-d milk yield were increased in parallel with the farm-scale and reached 8968.70 ± 124.56 kg and 7632.20 ± 79.67 kg, respectively, in the farms with the largest scale of 101 heads and above ($p < 0.001$). It was further determined that milk yield decreased significantly in the summer calving season compared to other seasons (Summer: 7897.20 ± 154.48 [b], Autumn: 8344.80 ± 169.33 [a], Winter: 8054.50 ± 127.22 [a], Spring: 8133.60 ± 159.77 [a]) ($p < 0.01$). Heat stress is thought to be the reason for the low milk yield in the summer season compared to other seasons. It was shown that the small-scale farms with 1–10 cows had the longest lactation length (394.90 ± 6.90 days) ($p < 0.001$). It was also determined that there is a directly proportional and significant relationship between the lactation number of Holstein cattle and lactation milk yield and 305-d milk yield values ($p < 0.01$). It was determined that dairy cattle in the 5th lactation had the highest 305-d milk yield value with 6992.00 ± 164.40 kg. In conclusion, a positive statistical correlation was found between the scale of dairy farms and their milk production parameter.

**Keywords:** Holstein-Friesian; farm-scale; heat stress; milk yield parameters

## 1. Introduction

In Turkey, the dairy cattle enterprises use a wide variety of operating, caring, and feeding methods. Milk yield parameters of these enterprises vary significantly according to the methods they use. While the conventional and unscientific operating, caring, and feeding methods are used in small farms, more scientific and modern methods are used in large dairy farms.

According to the data of Turkish Statistical Institute (TURKSTAT), in Turkey it is observed that the scale of 66.7% of cattle breeding enterprises is between 1 and 9 heads, and these enterprises have 18.4% of the bovine presence; the scale of 28.8% is between 10 and 49 heads and have 41.2% of the bovine presence; the scale of 3.9% is between 50 and 149 heads and have 20.9% of the bovine presence; the scale of 0.6% is over 150 heads and have 19.5% of the bovine presence [1].

Holstein Friesian is the most common cattle breed raised in Turkey, accounting for 65% [2]. Therefore, numerous studies are dealing with the effects of environmental factors on the milk yield traits of this breed [3–7]. However, studies on the interaction between farm-scale and milk yield are quite limited in Turkey [8].

The "Teke region" of Turkey is composed of Antalya, Burdur, Denizli, Isparta, and Muğla provinces. It was named after the "Teke Beylik", a Turkmen tribe that had settled in this region. The region possesses the characteristics of nomadic people who have had a unique social and cultural identity since the medieval time [9,10]. Animal husbandry has a special place in nomadic culture.

In the dairy sector, the productivity of cows is maximized within tight environmental conditions. If the temperature is either below or above the cow's comfort zone, economic efficiency and profitability are adversely affected. Heat stress negatively impacts a variety of productive traits, including milk yield [11].

Many studies have been conducted on the effect of lactation number on milk yield in cows. In many of these studies, it was stated that milk production increased with the number of lactations and peaked in the fourth or fifth lactation [12–14]. It is reported that the reason for this situation is due to the increase in the number of secretory cells owing to the increased development and size of the udder [15,16].

It is usually accepted that advancement and the adoption of contemporary and state-of-the-art technological innovation play an important role in enterprises' structural changes, which seems to be closely related to the concept of economies of scale [17–19] and minimizes the marginal and unit cost of enterprises and production, respectively [19–21]. It is noted that the spread of technology helps increase the sector output and leads to price decreases. The farmers who are reluctant to adopt new technologies remain deprived of the reward. It is necessary to have a high production size to provide technological innovations that increase profitability. In this context, large-scale enterprises are more advantageous in terms of technological innovation than small-scale enterprises [19]. Gloy et al. (2002) report that economies of scale are advantageous in dairy farms, and large-scale dairy holders who adopt higher-yielding production techniques tend to be more profitable than their smaller counterparts [22].

It has been stated in the different studies that milk yield can be used to evaluate regional technology differences and can be used to show structural changes in dairy cattle farms over time [23,24].

This study aims to determine the differences in terms of the milk yield characteristics of dairy cattle farms at different enterprise scales in Turkey. In order to minimize the effect of genetic and environmental factors on the results, the study is based on the data of the farms that raise pure Holstein Friesian cattle in the Teke Region of Turkey. Thus, an insight can be obtained about farm management, innovation, caring-feeding methods, and these methods' success among enterprises located in various scales.

## 2. Materials and Methods

In this study, the principles in the STROBE-Vet statement checklist [25] are considered. The material of the study consisted of 2005 of 12097 pieces of data, extracted from gaps and edge data, lactation records of 796 head of Holstein Friesian cows for the operating period between 2011 and 2017, which included the milk yield parameters of dairy farming enterprises in Antalya, Burdur, Denizli, and Isparta (Teke region) obtained from the inventory of the Cattle Breeders' Association of Turkey (CBAT).

In the research trial, the farm sizes were scaled according to the number of dairy cows, i.e., 1–10 heads, 11–50 heads, 51–100 heads, 101 heads, and above. The milk yield parameters included lactation length, 305 days adjusted milk yield (305-d milk yield), and total milk yield. The values of parameters were calculated according to the Trapezoidal rule in the Cattle Breeders' Association of Turkey's data evaluation system. As the number of lactations, each lactation between the 1st and 5th lactations was an individual group, whereas those from 6th to onwards lactations were categorized as a single lactation group. The calving seasons December, January, and February were classified as winter; March, April, and May as spring; June, July, and August as summer; and September, October, and November as autumn. Calving year was included in the scope of consecutive evaluation between 2011 and 2017 [14].

For heat stress interaction, by the calving season, average temperature and humidity values of the provinces' farms located were used to calculate the temperature humidity index (THI) for dairy [26] using TWC Product and Technology LLC, IBM database [27] in 2021.

The data were processed and statistically analyzed using Minitab® 16.1.1. General linear model analysis was employed to determine the relationship between milk yield characteristics and provinces, production scales of enterprises, lactation number, calving season, and calving year. Tukey's multiple comparison test was used to check significance of relationships between subgroups [13]. The THI values were also added the model but were excluded from the model for being statistically insignificant. For this purpose, below mentioned is the statistical model:

- $Y_{ijklm} = \mu + A_i + B_j + C_k + D_l + F_m + e_{ijklm}$
- $\mu$ = mean of total observed values
- $A_i$ = Effects of provinces (i = Antalya, Burdur, Denizli, Isparta)
- $B_j$ = Production scale effects of enterprises (j = 1–10, 11–50, 51–100, 101 head and above)
- $C_k$ = Effects of lactation number (k = 1,2,3,4,5,6 and above)
- $D_l$ = Seasonal effects (l = spring, summer, autumn, winter)
- $F_m$ = Effects of calving year (m = 2011,2012,2013,2014,2105,2016,2017)
- $Y_{ijklm}$ = Observed milk yield values at provinces i, scale of enterprises j, lactation number k, calving season l, calving year m

## 3. Results

The least square means for milk yield traits are presented in Table 1.

**Table 1.** The relationship between milk yield traits (Lactation length, 305-d milk yield, total milk yield) and environmental factors (provinces, production scales of enterprises, lactation number, calving season, and calving year) (Mean ± SE) (n = Holstein Friesian Cows).

| Factors | n | Lactation Length, d | 305-d Milk Yield, kg | Total Milk Yield, kg |
|---|---|---|---|---|
| **Province** | | | | |
| Antalya | 278 | 358.90 ± 6.19 | 5571.10 ± 117.18 c | 6726.50 ± 183.20 c |
| Burdur | 901 | 366.50 ± 4.21 | 6487.60 ± 79.68 b | 7797.10 ± 124.56 b |
| Denizli | 514 | 366.60 ± 4.91 | 8216.80 ± 92.86 a | 9934.00 ± 145.18 a |
| Isparta | 312 | 363.80 ± 6.45 | 6538.60 ± 122.11 b | 7972.40 ± 190.90 b |
| *p*-value | | 0.689 | 0.000 *** | 0.000 *** |
| **Farm-scale (head)** | | | | |
| 1–10 | 291 | 394.90 ± 6.90 a | 6099.60 ± 130.56 c | 7774.70 ± 204.11 c |
| 11–50 | 506 | 346.00 ± 4.98 c | 6192.90 ± 94.21 c | 7289.40 ± 147.28 c |
| 51–100 | 571 | 364.00 ± 4.93 b | 6889.50 ± 93.25 b | 8397.20 ± 145.78 b |
| ≥101 | 637 | 350.70 ± 4.21 bc | 7632.20 ± 79.67 a | 8968.70 ± 124.56 a |
| *p*-value | | 0.000 *** | 0.000 *** | 0.000 *** |
| **Lactation no.** | | | | |
| 1 | 603 | 361.40 ± 4.04 | 6338.70 ± 76.43 b | 7603.40 ± 119.49 c |
| 2 | 527 | 365.30 ± 4.48 | 6588.10 ± 84.85 ab | 7996.80 ± 132.65 abc |
| 3 | 372 | 356.90 ± 5.44 | 6648.00 ± 102.85 ab | 7851.00 ± 160.79 bc |
| 4 | 240 | 353.90 ± 6.76 | 6774.00 ± 127.91 ab | 7964.50 ± 199.96 abc |
| 5 | 141 | 361.50 ± 8.69 | 6992.00 ± 164.40 a | 8404.90 ± 257.02 ab |
| ≥6 | 122 | 384.60 ± 9.65 | 6880.20 ± 182.58 ab | 8824.60 ± 285.44 a |
| *p*-value | | 0.074 | 0.002 ** | 0.001 ** |

**Table 1.** *Cont.*

| Factors | n | Lactation Length, d | 305-d Milk Yield, kg | Total Milk Yield, kg |
|---|---|---|---|---|
| **Calving season** | | | | |
| Spring | 426 | 367.10 ± 5.40 ab | 6633.90 ± 102.20 ab | 8133.60 ± 159.77 a |
| Summer | 472 | 369.30 ± 5.22 a | 6458.10 ± 98.81 b | 7897.20 ± 154.48 b |
| Autumn | 351 | 365.10 ± 5.72 ab | 6899.60 ± 108.31 a | 8344.80 ± 169.33 a |
| Winter | 756 | 354.20 ± 4.30 b | 6822.60 ± 81.38 a | 8054.50 ± 127.22 a |
| *p*-value | | 0.026 * | 0.001 ** | 0.002 ** |
| **Calving year** | | | | |
| 2011 | 137 | 359.90 ± 9.08 | 6665.50 ± 171.82 bc | 7877.00 ± 268.61 abc |
| 2012 | 132 | 379.50 ± 9.12 | 6401.40 ± 172.58 c | 7975.00 ± 269.80 abc |
| 2013 | 256 | 369.50 ± 6.79 | 6495.90 ± 128.55 c | 7938.40 ± 200.96 bc |
| 2014 | 335 | 361.10 ± 5.90 | 6580.00 ± 111.61 bc | 7905.70 ± 174.49 bc |
| 2015 | 380 | 361.10 ± 5.40 | 6421.20 ± 102.10 c | 7836.90 ± 159.61 c |
| 2016 | 349 | 363.50 ± 5.43 | 6975.20 ± 102.70 b | 8486.00 ± 160.56 ab |
| 2017 | 416 | 352.90 ± 4.86 | 7385.50 ± 91.97 a | 8733.50 ± 143.79 a |
| *p*-value | | 0.179 | 0.000 *** | 0.000 *** |

*: $p < 0.05$, **: $p < 0.01$, ***: $p < 0.001$. abc: Means within the same column followed by different letters are statistically significant.

Average temperature and humidity values (by calving season), THI categories, and average THI values for calving seasons of the province centers are presented below (Tables 2 and 3).

**Table 2.** Average temperature and humidity values (by calving season) and THI categories of the province centers where the farms are located (Mean ± SE).

| THI Category | Value (F0) | Humidity (%) | n |
|---|---|---|---|
| No stress | 58.07 ± 0.21 | 67.08 ± 0.24 | 1468 |
| Mild stress | 77.48 ± 0.10 | 45.65 ± 0.40 | 435 |
| Severe stress | 82.59 ± 0.17 | 54.89 ± 0.54 | 102 |

Average THI (temperature humidity index) values.

**Table 3.** Average THI values for calving seasons in province centers (Mean ± SE).

| Antalya | | Burdur | | Denizli | | Isparta | |
|---|---|---|---|---|---|---|---|
| Spring | 68.92 ± 0.16 | Spring | 63.89 ± 0.09 | Spring | 63.87 ± 0.17 | Spring | 60.65 ± 0.14 |
| Summer | 83.82 ± 0.12 | Summer | 79.04 ± 0.04 | Summer | 79.40 ± 0.13 | Summer | 75.42 ± 0.06 |
| Autumn | 74.52 ± 0.12 | Autumn | 68.16 ± 0.12 | Autumn | 67.49 ± 0.17 | Autumn | 65.78 ± 0.24 |
| Winter | 60.02 ± 0.11 | Winter | 48.97 ± 0.10 | Winter | 50.97 ± 0.11 | Winter | 49.43 ± 0.24 |

Average THI (temperature humidity index) values.

## 4. Discussion

Although heat stress is an important factor in dairy depressing milk production [25], THI values were excluded from the model because they were statistically insignificant. In this study, THI values were calculated from the city centers' temperature and humidity values (Table 2). The climate of farm locations and the cities varies too wide, so this situation was suspected as the reason for statistical insignificancy. In future studies, the THI parameter could be added farm by farm.

There is no statistically significant difference between the lactation length recorded in the provinces (Table 1) ($p > 0.05$). However, statistically significant differences were found between the total milk yield and 305-d milk yield values of provinces ($p < 0.001$). It was found that Antalya had the lowest and Denizli had the highest milk yield, whereas Burdur and Isparta had similar milk yield values ($p < 0.001$). Since THI values of Antalya province

center were higher than other provinces, it is thought that heat stress negatively affects milk yield here more than other provinces and leads to lower milk production. In order to perform a more precise assessment in terms of heat stress, it is necessary to obtain climatic data of each farm. Although Burdur and Denizli province centers were nearly the same THI values, their town and villages had very different climatic conditions. The findings of the enterprises' values in Burdur province (305-d milk yield) were found to be above those obtained in different studies [28]. It is believed that the establishment of larger-scale modern enterprises in Burdur during this period increased the milk yield. In 2017, Holstein Friesian cattle's 305-d milk yield and total milk yield values in the Teke Region approached the value $7385.50 \pm 91.97$ kg, $8733.50 \pm 143.79$ kg respectively statistically significant ($p < 0.001$). As a variable, the year was considered as an important parameter from the viewpoint of milk yield in this study and many other literature studies [23,24]. While the value of milk yield results obtained in this study was higher than the value of the studies before the year 2000 in the Turkey, such as 305-d milk yield 4398 kg [29]; 305-d milk yield 5592 kg [30]; 305-d milk yield 4530.17 kg [31]; 305-d milk yield 4564.8 kg [32]; 305-d milk yield 4784 kg [33], the yield values of this study were found to be at average value of the studies performed after the year 2000 in the Turkey, such as 305-d milk yield 6884.11 kg [6], 305-d milk yield 7892.67 kg [7]. This situation suggested remarkable developments had been achieved in Holstein Fresian cattle enterprises of the Teke region in terms of technologies, breeding, genetics and farm management, caring, and feeding after the year 2000.

A considerable relationship was found between farm-scale and milk yield characteristics (Table 1) ($p < 0.001$). It had been observed that the lactation period was highest in value in farms with 1 to 10 heads of cattle. However, it was lowest in value in farms with 101 and above heads and 11–50 heads of cattle. This difference was found to be statistically significant ($p < 0.001$). Total milk yield and 305-d milk yield values were different considerably depending on the farm scale ($p < 0.001$). In this context, it was determined that the farms with 101 head of cattle and above have the highest value in terms of total milk yield and 305-d milk yield values ($p < 0.001$). In terms of total milk yield and 305-d milk yield, it was found that the enterprises with a scale of 1–10 and 11–50 heads have the lowest yield values. The enterprises in those two scales did not significantly differ in terms of the parameters, but the difference between them and larger-scale enterprises was extremely important ($p < 0.001$). These findings suggested that as the farm-scale grew, the enterprises were more likely to benefit from the advantages of economies of scale. These findings are in line with the results of previous studies [23,24] showing the relationship between farm scale and milk yield. In the light of the findings, as the scale of the enterprise grows, the success in management, organization, care, feeding, and breeding in dairy farms increases. Indeed, in a study using dairy cattle farm records in New England and New York, it was reported that there was a positive relationship between the farm-scale and technical efficiency [34]. Similarly, another study using dairy cattle farm records in New York suggested that the farm-scale has a positive effect on farm profitability [22]. These obtained findings were in sharp contrast to the findings of the study conducted by Galiç et al. (2004) in Izmir using the records of the CBAT for the period between 1996 and 2000 [8], which is the only study in the literature of Turkey on the relationship between farm-scale and milk yield parameters. One of the reasons for this is that the breeders in the Teke Region and the livestock enterprises of İzmir province have different characteristics in terms of socioeconomics. Another reason is that large-scale enterprises established with huge investment costs have gained awareness and enhanced their fund of knowledge regarding farm administration, caring, and feeding practices within the process.

While no significant relationship was found between the lactation number of Holstein cows and their lactation length ($p > 0.05$), a significant relationship was found between the total milk yield and 305-d milk yield values (Table 1) ($p < 0.01$). The data analysis indicated that as the number of lactations increased, the total milk yield and 305-d milk yield values also increased proportionally. It was determined that dairy cattle in the 5th lactation had the highest 305-d milk yield value with $6992.00 \pm 164.40$ and lowest value in

the first lactation with 6338.70 ± 76.43 kg. In terms of the relationship between lactation number and total milk yield and 305-d milk yield values, the results were similar to those of many other studies in the literature [7,32,35,36]. In many of the studies, it was stated that milk production increased with the number of lactations and peaked in the fourth or fifth lactation [12–14]. It is reported that the reason for this situation is due to the increase in the number of secretory cells owing to the increased development and size of the udder [15,16]. Furthermore, less mature cows have a different endocrine background that limits the separation of nutrients into milk at birth [37]. In addition, the daily feed consumption of primiparous cows is lower than that of multiparous cows [38] and the body size of older cows increases compared to animals in the first lactation [14].

A significant relationship was found between the calving season and milk yield parameters (Table 1). It was observed that the cows that calved in the summer season had the longest lactation length as compared to the cows which calved in the winter season ($p < 0.05$). The total milk yield and 305-d milk yield parameters further determined that milk yield decreased significantly in the summer season compared to other seasons ($p < 0.01$). Similarly, many other studies suggest that there is a relationship between calving season and both total milk yield and 305-d milk yield parameters [35,36,39,40]. The heat stress was believed the main reason for the low milk yield during summer season. Many studies put forward suggestions that heat-stress reduces animals feed intake and this inefficient feed intake is responsible for decreased milk yield [11,41,42].

No significant relationship was found between calving year and lactation length (Table 1) ($p > 0.05$). However, the relationship between calving year and total milk yield and 305-d milk yield values was found to be statistically significant ($p < 0.001$). Particularly, in 2017, the total milk yield and 305-d milk yield values were considerably higher than those in other years. While the relationship between calving year, total milk yield and 305-d milk yield values was found to be significant in numerous studies [31,32,35,36], the relationship between calving year and 305-d milk yield according to some studies [6,28] and between calving season and total milk yield according to some other studies is not significant [4].

## 5. Conclusions

This study's results are important in that they indicate a statistically significant relationship between farm-scale and milk yield in the Teke Region. As the farm-scale grows, the profitability and efficiency of the farm increases. Moreover, the power of providing technological innovations in matters, such as farm management, caring, and feeding methods, is increasing as well. This is reflected as the scale-related increase in milk productivity, which is the most important income item of dairy cattle farms. The reason for the decrease in milk yield in summer calving season, which is statistically significant (in Table 1), needs to be researched deeply. There is a need for more detailed studies on the effects of heat stress in the region.

It is observed that the milk yield values of Holstein Friesian cattle in the Teke Region are somewhat above the milk yield values of Holstein Friesian cattle of the Teke region reported by almost all of the studies conducted before 2000. This supports that considerable improvements have been achieved in the technologies, animal improvement, genetics, farm administration, caring, and feeding practices of Holstein Friesian cattle breeders in the region since the 2000.

**Funding:** This research received no external funding.

**Institutional Review Board Statement:** The animal study protocol was approved by Ethics Committee) of Non-Invasive Clinical Research Ethics Committee of Mehmet Akif Ersoy University (GO 2018/102 and 10.03.2018).

**Informed Consent Statement:** Not applicable.

**Data Availability Statement:** The data presented in this study are available on request from the corresponding author.

**Conflicts of Interest:** The authors declare no conflict of interest.

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
