# Peer review of "The Effect of Different Environmental Factors on Milk Yield Characteristics of Holstein Fresian Cattle Raised with Different Production Scale on Teke Region of Turkey"

_sustainability, doi:10.3390/su142113802_

Round 1

Reviewer 1 Report

Article 1 Effect of Farm-Scale, Heat Stress and Different Methods on Milk Yield Parameters of Holstein Friesian in the Teke Region of Turkey

Keywords: Holstein-Friesian; farm-scale; heat stress; milk yield parameters

Some words appear in the title and keywords simultaneously. You need to get it out of one. No titles or keywords should appear. Can't repeat.

Why wasn't the bull effect introduced in the herd?

Do owners use natural mating or artificial insemination?

This information needs to be described in the material and method.

How many bulls do they use per year?

Do they make selections? What's the criterion?

…..

the statistical model: • Yijklm= µ+ Ai+Bj+Ck+Dl+Fm+eijklm

What was the average calving interval for the cows?

In table 2 and 3, the must be described in the footer of the table

Table 3. Average THI [temperature humidity index (THI)] values …

Why not use a more robust model, nesting the THI effect within provinces? Why wasn't the tour effect inserted in the model?

Author Response

Response to Reviewer 1 Comments

Point 1: Some words appear in the title and keywords simultaneously. You need to get it out of one. No titles or keywords should appear. Can't repeat.

Response 1: Dear Reviewer, I found your suggestion very appropriate and I revised the title.as "The Effect of Different Environmental Factors on Milk Yield Characteristics of Holstein Fresian Cattle Raised with Different Production Scale on Teke Region of Turkey". I removed the term Holstein Fresian from the keywords.

Point 2: Why wasn't the bull effect introduced in the herd?

Do owners use natural mating or artificial insemination?

This information needs to be described in the material and method.

How many bulls do they use per year?

Do they make selections? What's the criterion?

What was the average calving interval for the cows?

Response 2: Dear Reviewer, the effect of environmental factors on milk yield traits was investigated in this study. Its effect on fertility traits has not been my focus. So, I think that the topics you mentioned about fertility do not need to be included in this study.

Point 3: In table 2 and 3, the must be described in the footer of the table. Table 3. Average THI [temperature humidity index (THI)] values ….

Response 3: Dear Reviewer, footer has been added to Table 2 and Table 3 as you mentioned in your suggestion.

Point 4: Why not use a more robust model, nesting the THI effect within provinces? Why wasn't the tour effect inserted in the model?

Response 4: Dear Reviewer, you can understand that my main aims to determine the differences in terms of the milk yield characteristics of dairy cattle farms at different enterprise scales in Turkey. When I sent this article to a journal for the first time, a valuable reviewer stated that the decrease in milk yield in summer may be due to THI. He added that sharing THI information would enrich the publication. I liked the reviewer's suggestion very much and I added THI in 2021 as an addendum. However, I think, as I mentioned in the publication (L128-129), the data I could access did not fully reflect the results in the villages, since they belonged to the city center. I could not access the weather data of the villages of the provinces I visited in any database. I shared this openly. When I included it in the model, it was breaking the integrity of the model. However, I did not remove this data from the publication, as I thought it provided a perspective for the publication. Dear reviewer, unfortunately, I could not reach tour effect as these data were obtained from the Cattle Breeders' Association of Turkey database.  

Best regards.

Reviewer 2 Report

* Effect of Farm-Scale, Heat Stress and Different Methods on Milk Yield Parameters of Holstein Friesian in the Teke Region of Turkey

Dear author,

Thank you very much for allow me to read this interesting manuscript.

I think that the most interesting point in this study is the richness of the data.
With that information you could make interesting analyses and suggest actions for
the dairy industry to improve the all process of dairy production.

I suggest you make the data available in order to make this research more
visible, improve the repeatability of the study and to facilitate future studies
(e.g., meta-analysis).

I consider that the manuscript can be improved, mainly by correcting some writing
issues and improving the data processing description. I have multiple questions about
how the data was extracted, processed and analysed. I suggest that you read the STROBE
veterinary statement to have a reference to report descriptive studies.

I offer the following comments and hope they can be useful to improve the
manuscript.

** Title

 * What is the meaning of "different methods" on the sentence?
 * Please check it and reword, if necessary.

** Abstract

L 11, L 14:

I think that the following expressions does not add meaning to the sentences:

 * "The analyzes showed that" (L 11)
 * "The total milk yield and 305-d milk yield parameters further determined that" (L 14)

Please check and rewrite.

L 15 and elsewhere:

Please include the value to allow the readers have an idea of the strength of
the association. Note that P-values are meaningless when used without further
references such as means, type of models and the number of observations used
to estimate the confidence intervals.

L 16:

Please check the wording of this sentence.

L 17:

- I suggest you remove the expression "head of"

**  Introduction

L 28:

- I suggest you remove the expression "It is observed in Turkey that"

L 70-71:

- Previoulsy you mentioned that "heat stress" as well as "different methods" were
  also assessed

** Materials and Methods

L77:

- Did you used repeated measurements within the same unit of analyses?
  e.g., were there repeated measurements for the same cow or same farm throughout
   this period?
- This may suggest that there are pseudoreplicated measurements. Please describe
  this in the manuscript: whether or not existed pseudoreplicated measurements and
  how did you deal with those measurements to make your analyses correspond with
   the appropriate degrees of freedom.

L 79:

- Did you collect data on milk quality?
  - It would be valuable to adjust the milk yield by solids, or use fat corrected
     milk production, as it will make comparisons more fairer.

L 82-83:

- How did you arrive at these categories?

L 93-94:

- Please inform the source of these data.
- Please inform the time lag used to calculate the THI:
  - Did you use data collected at each hour, day?
- The THI is proxy for variables that directly affects the individual and, most
  of the times, it is related with the time of the day. Thus, the use of that
  indicators makes more sense when you are running cow-level analyses and when
  the weather is measured within small units (perhaps minutes). Simple averages
  of temperature and humidity may hide important information, since most of the
  times the mean is not representative of this asymmetric variable.
- According to the previous idea, your analyses may be biased towards something
  similar to an "ecological fallacy". Do you have any thoughts about this?

L 96:

Since the entire manuscript is based on extracting and processing databases I think
you should describe in-deep:

- The process of data extraction, cleaning and processing before performing
  statistical analyses
- How did you deal with any abnormal entry or missing data?
- Would you make all the data and scripts available to make your study repeatable?

L 97:

- I am not sure about the meaning of the expression "General Linear Model".
  - Did you use a simple linear regression model?
  - Or did you use a GLM with specific error distribution? (e.g., gamma, beta...)
  - Did you model the data using mixed-effects models?
- Please inform if you checked the goodness of fit of the models and what methods
  did you use.    
- How did you account for potential confounders or intervening variables?

L 102:

- Please inform the unit of analysis:
  - Did you perform: cow, herd, or province-level analyses?
  - In any of these options could be considerations regarding hierarchical data.
    Do you think it should be used any statistical tool to account for variance
    components allocated to each one of the levels?
  - Did you identity any pseudoreplicated measurement in your data?
    - How did you deal with that situation?

** Results

L 113:

- I think that to include some description of the sample could be useful for the
  readers deeply understand your manuscript. Most of the variables you collected
  are asymmetric, thus medians and IQR are very important to asses the data
  distribution.
- Why is there no description (sentences on the manuscript) of the results you are
  presenting in the Tables?
- Did you collect data to characterize each farm type or production units?
  - Were the cows housed or managed in grazing systems?
  - How were the cows managed?
  - What kind of feed is commonly offered to the cows?
  - What type of reproductive management was commonly used by the farms?
  - Did the farms have structured management, sanitary, reproductive and nutritional
    programs?
  - Did the farms have technical assistance by veterinarians, nutritionist or any
    other professional?

L 115 and elsewhere:

- Please improve the title of the table. It should be more descriptive and inform
  the readers about the data you are presenting.
- What "n" means on this Table?
- I think that you put too much information in one Table.
  - Do you think it is possible to split the information to improve the readability?

** Discussion

L 133-137;  158-168; L 186-193; 202-207:

- These sentences seems more like a description of the results.
 - Did you intend to include both Results and Discussion on the same section
   of the manuscript?

L 195-196:

- Some high-performance dairy Holstein cows have their large dairy production at
  the third lactation:
  - Do you think that is common?
  - What is the factor that could explain this?
  - Is there any association between longevity and 305 day adjusted milk yield
     in Holstein?

** Conclusions

L 224-225:

- Do you have data, or any description of the sample of farms, to support this
  statement?

L 228:

- However you used an indicator of heat stress in dairy cows and there was not
  association?
 - Could you elaborate on this issue? (any consideration to include in the
   manuscript should go to the Discussion)

Author Response

Response to Reviewer 2 Comments

Point 1: What is the meaning of "different methods" on the sentence? Please check it and reword, if necessary.

Response 1: Dear Reviewer, in Turkey, there is a wide range of dairy cattle enterprises from family type small-scale farms using conventional techniques to large-scale farms engaged in intensive production. Revised the title as "The Effect of Different Environmental Factors on Milk Yield Characteristics of Holstein Fresian Cattle Raised with Different Production Scale on Teke Region of Turkey". English is not my native language. I am open to your suggestions.

Point 2: I think that the following expressions does not add meaning to the sentences:

* "The analyzes showed that" (L 11)

* "The total milk yield and 305-d milk yield parameters further determined that" (L 14)

Please check and rewrite.

Response 2: Dear Reviewer, "The analyzes showed that" (L 11) removed. "The total milk yield and 305-d milk yield parameters further determined that" (L 14) expression revised as to “It further determined".

Point 3: L 15 and elsewhere: Please include the value to allow the readers have an idea of the strength of the association. Note that P-values are meaningless when used without further references such as means, type of models and the number of observations use to estimate the confidence intervals.

Response 3: Dear Reviewer, milk yield parameters added. (Summer: 7897.20±154.48b, Autumn: 8344.80±169.33a, Winter: 8054.50±127.22a, Spring: 8133.60±159.77a)(P<0.01).

Point 4: L 16: Please check the wording of this sentence..

Response 4: Dear Reviewer, L16 is revised as: “Heat stress is thought to be the reason for the low milk yield in the summer season compared to other seasons.”

Point 5: L 17:I suggest you remove the expression "head of"

Response 5: Dear Reviewer, L 17 “head of” removed.

Point 6: L 28: - I suggest you remove the expression "It is observed in Turkey that"

Response 6: Dear Reviewer, "It is observed in Turkey that" (L 28) removed.

Point 7: - Previoulsy you mentioned that "heat stress" as well as "different methods" were also assessed

Response 7: Dear Reviewer, I could not express well what I wanted to give in the content with the expression I used in the title. That's why I thought it appropriate to change the title as "Milk Yield Characteristics of Holstein Fresian Cattle Raised with Different Production Scale and Production Method from Traditional to Intensive in the Teke Region of Turkey".

Point 8: L 28: - I suggest you remove the expression "It is observed in Turkey that"

Response 8: Dear Reviewer, "It is observed in Turkey that" (L 28) removed.

Point 9: - Did you used repeated measurements within the same unit of analyses? e.g., were there repeated measurements for the same cow or same farm throughout this period?- This may suggest that there are pseudoreplicated measurements. Please describe this in the manuscript: whether or not existed pseudoreplicated measurements and how did you deal with those measurements to make your analyses correspond with the appropriate degrees of freedom.

Response 9: Dear Reviewer, as a general rule, it is more efficient to have more experimental units with fewer samples per unit than fewer units with more samples. So I manage to study on this rule. I tried to provide samples from as many different farms as possible.Yes, I used repeated measurements within the same unit of analyses. It’s the nature of this study. In practice, values of 70% and 80% are commonly used in field trials as the desired level of statistical power. Many field trials study NTOs, with separate fields as replicates. Therefore, large numbers of replicates are needed over several seasons to test the hypotheses in the face of effect of confounding environmental variables. A power analysis indicated that replication of 20 experimental units (fields) per crop per year over 3 years (in total, 60 replicates) should yield adequate power (>80%) to detect differences of 1.5 fold or to detect 50% difference (Perry et al. 2003). I also got data from 796 cows on average of 3 lactations. Thus, I tried to increase both different business areas and statistical power.

Point 10: - Did you collect data on milk quality? - It would be valuable to adjust the milk yield by solids, or use fat corrected milk production, as it will make comparisons more fairer.

Response 10: Dear Reviewer, unfortunately, I could not analyze milk fat and dry matter, as I could not provide adequate financial resources.

Point 11: - L 82-83: - How did you arrive at these categories?

Response 11: Dear Reviewer, the reason for the selection of 1-10 heads is that farms up to this scale cannot receive incentives in Turkey. Farms up to this scale make traditional production. Their priority is to meet their own milk and dairy products and meat needs. Livestock production is not the main source of income for these enterprises and generally supports crop production. On the other hand, farms of 100 heads or more are businesses that have all kinds of resources for competitive and innovative livestock production (e.g, mechanization required for production and having fields for the production of forage crops), in contrast to 1-10 head farms. The most basic selection criterion in the selection of 51-100 head farms is having a milking unit, which is an important innovation in dairy cattle breeding. Farms with 11-50 milking cows are enterprises that are in the development stage with similar characteristics, barely standing, and innovatively lacking some important features (no milking units, not have adequate farms..).

Point 12: L 93-94: - Please inform the source of these data. - Please inform the time lag used to calculate the THI: - Did you use data collected at each hour, day? - The THI is proxy for variables that directly affects the individual and, most  of the times, it is related with the time of the day. Thus, the use of that  indicators makes more sense when you are running cow-level analyses and when  the weather is measured within small units (perhaps minutes). Simple averages  of temperature and humidity may hide important information, since most of the times the mean is not representative of this asymmetric variable. - According to the previous idea, your analyses may be biased towards something similar to an "ecological fallacy". Do you have any thoughts about this?

Response 12: Dear Reviewer, I informed the source of these data the time lag used to calculate the THI as “…using TWC Product and Technology LLC, IBM database [41] in 2021.” You can understand that my main aims to determine the differences in terms of the milk yield characteristics of dairy cattle farms at different enterprise scales in Turkey. When I sent this article to a journal for the first time, a valuable reviewer stated that the decrease in milk yield in summer may be due to THI. He added that sharing THI information would enrich the publication. I liked the reviewer's suggestion very much and I added THI in 2021 as an addendum. However, I think, as I mentioned in the publication (L128-129), the data I could access did not fully reflect the results in the villages, since they belonged to the city center. I could not access the weather data of the villages of the provinces I visited in any database. I shared this openly. However, I did not remove this data from the publication, as I thought it provided a perspective for the publication. I value your opinion and suggestion about this context.

Point 13: L 96: Since the entire manuscript is based on extracting and processing databases I think

you should describe in-deep:

- The process of data extraction, cleaning and processing before performing statistical analyses. - How did you deal with any abnormal entry or missing data? - Would you make all the data and scripts available to make your study repeatable?

Response 13: Dear Reviewer, initially, there were 12097 pieces of lactation records of Holstein Friesian cows. First of all, from these records, those with spaces in the data set were removed. The extreme data were then excluded from the study. After all, the data were again examined in terms of the general nature of dairy cattle and materials containing illogical data were removed. After all this sorting process, analyzes were performed with the remaining 2005 data. L 77: The sentence revised at “The material of the study consisted of 2005 of 12097 pieces of data, extracted from gaps and edge data,…” I am sending you the raw and processed data and the results of the GLM analysis made in the Minitab program. I will be pleased with any contribution you to improve the manuscript.

Point 14: L 97:- I am not sure about the meaning of the expression "General Linear Model". - Did you use a simple linear regression model? - Or did you use a GLM with specific error distribution? (e.g., gamma, beta...) - Did you model the data using mixed-effects models? - Please inform if you checked the goodness of fit of the models and what methods did you use. - How did you account for potential confounders or intervening variables?

Response 14: Dear Reviewer, if you review the raw data and analysis results I sent you and let me know what's missing, I will try to correct it as much as I can. I can only say that I did the work with the least squares method in the GLM analysis, which is frequently used in field studies in Turkey. It is reported that this model is used in studies in our country because it automatically sorts out extreme and repeated measured data.

Point 15: L 102: - Please inform the unit of analysis: - Did you perform: cow, herd, or province-level analyses?  - In any of these options could be considerations regarding hierarchical data. Do you think it should be used any statistical tool to account for variance. Components allocated to each one of the levels? - Did you identity any pseudoreplicated measurement in your data?- How did you deal with that situation?

Response 15: Dear Reviewer, I performed all the items in the model by explained in the model. You can examine the answers to your questions on this matter from the data I have forwarded to you and the Minitab analysis.

Point 16: L 113:- I think that to include some description of the sample could be useful for the readers deeply understand your manuscript. Most of the variables you collected are asymmetric, thus medians and IQR are very important to asses the data distribution. - Why is there no description (sentences on the manuscript) of the results you are  presenting in the Tables?- Did you collect data to characterize each farm type or production units?

  - Were the cows housed or managed in grazing systems?

  - How were the cows managed?

  - What kind of feed is commonly offered to the cows?

  - What type of reproductive management was commonly used by the farms?

  - Did the farms have structured management, sanitary, reproductive and nutritional

    programs?

  - Did the farms have technical assistance by veterinarians, nutritionist or any

    other professional?

Response 16: Dear Reviewer, in the article, I presented the data with standard error values. This is also given in many studies in the literature. If I had given it as you stated, unfortunately I would have had a very difficult time finding a journal to publish. Since most of the journals limit the publications to 5 pages and the summary to 250 words, I have to summarize as much as possible in the publications. An important part of the questions you ask in this title are the elements that are not mentioned in the concept of the publication (Were the cows housed or managed in grazing systems?

  - How were the cows managed?

  - What kind of feed is commonly offered to the cows?

  - What type of reproductive management was commonly used by the farms?

  - Did the farms have structured management, sanitary, reproductive and nutritional

    programs?

  - Did the farms have technical assistance by veterinarians, nutritionist or any

    other professional?).

Point 17: L 115 and elsewhere: - Please improve the title of the table. It should be more descriptive and inform the readers about the data you are presenting. - What "n" means on this Table?

- I think that you put too much information in one Table. - Do you think it is possible to split the information to improve the readability?

Response 17: Dear Reviewer, title of Table 1 revised as “The relationship between milk yield traits (Lactation length, 305-d milk yield, total milk yield) and environmental factors (provinces, production scales of enterprises, lactation number, calv-ing season, and calving year) (Mean±SE) (n= Holstein Friesian Cows)”.

Point 18: L 133-137; 158-168; L 186-193; 202-207: - These sentences seems more like a description of the results. - Did you intend to include both Results and Discussion on the same section of the manuscript?

Response 18: Dear Reviewer, L 133-137; 158-168; L 186-193; 202-207: Yes, I intend to both Results and Discussion on the same section of the manuscript. But unfortunatelly, there is no information that it can be written in the concept you specified in the journal writing guide. 

Point 19: L 195-196: - Some high-performance dairy Holstein cows have their large dairy production at the third lactation: - Do you think that is common? - What is the factor that could explain this?

- Is there any association between longevity and 305 day adjusted milk yield in Holstein??

Response 19: Dear Reviewer, That is not common in this study. According to my observation, insufficient feeding from birth to all longevity, causes the lactation peak to be formed later than the third lactation in Turkey.

Point 20: L 224-225:- Do you have data, or any description of the sample of farms, to support this statement?

Response 20: Dear Reviewer, as I mentioned Response 11, the mechanization (milking unit, feed mixing machine, etc.) and production capacity of their own feeds are much higher in farms with large production scales.

Point 21: L 228:- However you used an indicator of heat stress in dairy cows and there was not association? - Could you elaborate on this issue? (any consideration to include in the manuscript should go to the Discussion)

Response 21: Dear Reviewer, I believe heat stress and milk yield was negative correlation relationship. But as I mentioned in L 127-132, the climate of farms location and the cities were varies too wide so this situation was suspected as the reason for statistically insignificancy.

Note: The system is not given permission me to sent you my database in upload menu. They are ready but I can not sent to you sir/madam.

Kind regards.

Reviewer 3 Report

The manuscript entitled Effect of Farm-Scale, Heat Stress and Different Methods on Milk Yield Parameters of Holstein Friesian in the Teke Region of Turkey. Here in this article, the author determined whether there was any difference between the parameters of herd size and milk yield based on hypothesis that the dairy cattle enterprises in the Teke Region used different production methods depending on their herd size. This article is interesting and worth to be published, although I listed a few items below for the authors considerations. The research manuscript needs revisions and improvements prior to publication processing.

Comment 1. In the title, author mention about the effect of heat stress and different method, but the heat stress parameter itself haven’t been explained in the description, how to determine heat stress?  

Comment 2. What is different method means stated in the title? What method? What is the difference. Please explained more in detail

Comment 3.  How about the feeding strategy in each farm scale? could be better if there are the data of feed composition they use and how is the feeding management. Please mention briefly and perhaps there also correlate with the results. 

Author Response

Response to Reviewer 3 Comments

Point 1: In the title, author mention about the effect of heat stress and different method, but the heat stress parameter itself haven’t been explained in the description, how to determine heat stress?

Response 1: Dear reviewer, I found your suggestion very appropriate and I revised the title.as "The Effect of Different Environmental Factors on Milk Yield Characteristics of Holstein Fresian Cattle Raised with Different Production Scale on Teke Region of Turkey". English is not my native language. I am open to your suggestions.

Point 2: What is different method means stated in the title? What method? What is the difference. Please explained more in detail.

Response 2: Dear reviewer, I realized that I couldn't express what I wanted to say well in the title because English is not my native language. To solve all these problems, I wrote a title that I think fits the content much better. I'm open to your comments on the title.

Point 3: How about the feeding strategy in each farm scale? could be better if there are the data of feed composition they use and how is the feeding management. Please mention briefly and perhaps there also correlate with the results.

Response 3: Dear reviewer, unfortunately, I could not reach feeding strategy as these data were obtained from the Cattle Breeders' Association of Turkey database. However, as I mentioned in the article, I used the scale of the enterprise as an indicator of better feeding and maintenance, more innovative production, more land availability for forage cultivation.

Kind regards.

Round 2

Reviewer 2 Report

The Effect of Different Environmental Factors on Milk Yield 2
Characteristics of Holstein Fresian Cattle Raised with Different 3
Production Scale on Teke Region of Turkey

Dear author,

Thank you very much for your response.

** Please check the STROBE veterinary statemet

I strongly recommend that you read and check the STROBE veterinary statement, as I suggested in the first round:

- https://www.sciencedirect.com/science/article/pii/S0167587716303282?via%3Dihub

Please note that a checklist is availiable, so you can check your entire manuscript for any information that perhaps is lacking.

Most of my comments are related to adjust your manuscript to the recommendations of the Strobe Veterinary Statement and to accomplish with good practices in data management reporting to make clear to your readers
that your study is repeatable.

** Some comments were not addressed

I believe I have not been so clear with some of my comments and that could have lead to a misunderstanding.

The following comments were not completely addressed, or some questions were not asked (Point 14, Point 15, Point 16, Point 21).

** There is a contradiction between a statement in the conclusion and your results

Please note that in L 228 you are concluding with a sentence that seems to contradict the study results:

"It is believed that the reason for the decrease in milk yield in summer calving season, which is statistically significant, 232
is heat stress."

Note that in L 31-33 you said that:

"Although the heat stress is an important factor in dairy for depressing milk produc- 131
tion [25], THI values were excluded from the model because they were statistically insig- 132
nificant."

** About publishing you data and code

Regarding the data and code, my suggestions were not intended to make a revision of your code or data. Instead, my suggestions aimed at promoting open science. You can make available data and code in external repositories like Zenodo, in order to your readers have access to them.

Publishing data and code may be good idea in order to:
 - Increase the visibility and relevance of your study
 - Improve the understanding of your studies by the public
 - Improve the reproducibility (https://en.wikipedia.org/wiki/Reproducibility)
   and replicability (https://www.ncbi.nlm.nih.gov/books/NBK547525/) of your study
 - Be committed with open science
 - Please check https://molecularbrain.biomedcentral.com/articles/10.1186/s13041-020-0552-2 for
   additional information about this issue

Author Response

Point 1: Please check the STROBE veterinary statemet

I strongly recommend that you read and check the STROBE veterinary statement, as I suggested in the first round:

- https://www.sciencedirect.com/science/article/pii/S0167587716303282?via%3Dihub

Please note that a checklist is availiable, so you can check your entire manuscript for any information that perhaps is lacking.

Most of my comments are related to adjust your manuscript to the recommendations of the Strobe Veterinary Statement and to accomplish with good practices in data management reporting to make clear to your readers that your study is repeatable.

Response 1: Dear Reviewer, I want you to know that I really care about your reviews. Even your guidance with your comments and especially the “Strobe Vet Statement” has broadened my vision of my profession. Really thank you so much. I take the “Strobe Vet Statement” seriously and will direct the planning of my future field studies according to this statement. I can honestly say that the content of STROBE will create a very solid framework for my field area.

If you let me, I want to add this sentence in material and method: “In this study, the principles in the STROBE-Vet statement checklist are considered.”(L 78).

Point 2: ** There is a contradiction between a statement in the conclusion and your results

Please note that in L 228 you are concluding with a sentence that seems to contradict the study results:

"It is believed that the reason for the decrease in milk yield in summer calving season, which is statistically significant is heat stress."

Note that in L 31-33 you said that:

"Although the heat stress is an important factor in dairy for depressing milk production [25], THI values were excluded from the model because they were statistically insignificant."

Response 2: Dear Reviewer, L232 is revised as: " The reason for the decrease in milk yield in summer calving season, which is statistically significant (in Table 1), is needed to be researched deeply.” 

Point 3: About publishing you data and code

Regarding the data and code, my suggestions were not intended to make a revision of your code or data. Instead, my suggestions aimed at promoting open science. You can make available data and code in external repositories like Zenodo, in order to your readers have access to them.

Publishing data and code may be good idea in order to:

 - Increase the visibility and relevance of your study

 - Improve the understanding of your studies by the public

 - Improve the reproducibility (https://en.wikipedia.org/wiki/Reproducibility)

   and replicability (https://www.ncbi.nlm.nih.gov/books/NBK547525/) of your study

 - Be committed with open science

 - Please check https://molecularbrain.biomedcentral.com/articles/10.1186/s13041-020-0552-2 for   additional information about this issue.

Response 3: Dear Reviewer, I wholeheartedly agree with your opinions on open data sharing. I would like to state that I have read with great interest the web site addresses you sent for my perspective development. However, in my country, unfortunately, it is not possible to do anything freely in the last twenty years, and to do science. If you are against the government, if you are open-minded, secular, accepting science as a guide, and a defender of freedom and justice, you are punished in my country. For these reasons, although I would like to share the raw data with everyone as I shared with you, I cannot do this due to the conditions of my country, forgive me.

Kind regards.
